# The Transcription Factor MbWRKY46 in *Malus baccata* (L.) Borkh Mediate Cold and Drought Stress Responses

**DOI:** 10.3390/ijms241512468

**Published:** 2023-08-05

**Authors:** Wanda Liu, Tianhe Wang, Yu Wang, Xiaoqi Liang, Jilong Han, Ruining Hou, Deguo Han

**Affiliations:** 1Horticulture Branch, Heilongjiang Academy of Agricultural Sciences, Harbin 150040, China; haaswth@126.com (T.W.); haaswangyu@126.com (Y.W.); hanjilong0000@163.com (J.H.); hlj6658@163.com (R.H.); 2Key Laboratory of Biology and Genetic Improvement of Horticultural Crops (Northeast Region), Ministry of Agriculture and Rural Affairs, National-Local Joint Engineering Research Center for Development and Utilization of Small Fruits in Cold Regions, College of Horticulture & Landscape Architecture, Northeast Agricultural University, Harbin 150030, China; a13989332297@163.com

**Keywords:** *M. baccata* (L.) Borkh., *MbWRKY46*, cold stress, drought stress, transgenic plant

## Abstract

The living environment of plants is not static; as such, they will inevitably be threatened by various external factors for their growth and development. In order to ensure the healthy growth of plants, in addition to artificial interference, the most important and effective method is to rely on the role of transcription factors in the regulatory network of plant responses to abiotic stress. This study conducted bioinformatics analysis on the *MbWRKY46* gene, which was obtained through gene cloning technology from *Malus baccata* (L.) Borkh, and found that the *MbWRKY46* gene had a total length of 1068 bp and encodes 355 amino acids. The theoretical molecular weight (MW) of the MbWRKY46 protein was 39.76 kDa, the theoretical isoelectric point (pI) was 5.55, and the average hydrophilicity coefficient was −0.824. The subcellular localization results showed that it was located in the nucleus. After conducting stress resistance studies on it, it was found that the expression of *MbWRKY46* was tissue specific, with the highest expression level in roots and old leaves. Low temperature and drought had a stronger induction effect on the expression of this gene. Under low temperature and drought treatment, the expression levels of several downstream genes related to low temperature and drought stress (*AtKIN1*, *AtRD29A*, *AtCOR47A*, *AtDREB2A*, *AtERD10*, *AtRD29B*) increased more significantly in transgenic *Arabidopsis*. This indicated that *MbWRKY46* gene can be induced to upregulate expression in *Arabidopsis* under cold and water deficient environments. The results of this study have a certain reference value for the application of *M. baccata MbWRKY46* in low-temperature and drought response, and provide a theoretical basis for further research on its function in the future.

## 1. Introduction

Apple is widely planted in over 90 countries and regions around the world [1]. China is the largest apple producing country, especially in the northern region, where it is widely planted. However, the changing environment can easily affect the quality and yield of apples [2]. Apples are propagated using grafting technology, so their stress resistance mainly depends on the rootstock. *Malus baccata* (L.) Borkh. has strong cold resistance and is the main apple rootstock in Northeast and North China [3,4]. It has good application prospects in cold resistance breeding, multi resistance rootstock, processing, and landscape. Cold stress can be divided into two forms: chilling (0–15 °C) and freezing (<0 °C) [5]. If plants are exposed to low-temperature environmental conditions for an extended time, their growth and development will be inhibited, and when the temperature is below 0 °C, there is a high possibility of cell membrane damage leading to cell death [6]. When plants are under other stress conditions, including drought, salinity, high temperature and Abscisic acid (ABA), their growth and development also face significant threats [7,8]. In the process of evolution, plants have formed a complex signaling mechanism that adapts to stress and regulates related physiological and biochemical indicators. In this regulatory mechanism network, transcription factors (TFs) play a crucial role in binding to eukaryotic gene promoter specific sites to activate or inhibit the expression of target genes [9].

WRKY TFs are unique to plants, which have many members and are one of the largest families in the TF family. WRKY TFs have a conserved amino acid sequence at the N-terminus (WRKYGQK, which is unique to WRKY TFs) and a conserved zinc finger structure at the C-terminus (Cx4-5Cx22-23HxH or Cx7Cx23HxC) [10]. Different WRKY TFs have different number of domains and types of zinc finger motif Order type; based on this, WRKY proteins can be divided into three categories. Those with two domains and a zinc finger structure of CX 4 C22-23 HXH are group I; the group with only one domain and zinc finger structure of CX 4-5 C 23 HXH is Group II [11]; the group with one domain and zinc finger structure of C2-H-C (C-X7-C-X23-H-X1-C) is Group III [10]. Research has shown that WRKY TF is widely involved in plant growth, aging, and metabolic activities, regulating plant responses to various stresses [12,13]. Yu et al. characterized the drought resistance function of wheat *TaWRKY1*-2D and found that this gene was an important candidate gene for plant response to drought stress [14]. It was found that drought and ABA can strongly induce the expression of PheWRKY86, and that *PheWRKY86* can improve plant tolerance by regulating the expression of *NCED1* [15]. Cucumber CsWRKY46 increased the expression of downstream stress induced genes such as *RD29A* and *COR47* through ABA dependent pathways, endowing transgenic plants with cold resistance [16].

Since the first isolation of WRKY TF from sweet potato (*Ipomoea batatas*) and the discovery that its expression can be induced by sucrose [17], research on WRKY has continued to deepen. Significant results have been achieved in the study of many WRKY TFs isolated from and identified in various plants such as *Arabidopsis*, rice, cucumber, and corn [16,18,19,20]. However, research on *Malus* WRKY TF is not yet deep enough. The apple industry is one of the most important parts of the Chinese agricultural industry, but various environmental factors (such as low temperature, drought, salinity, etc.) threaten the growth of apple trees, leading to the inability to guarantee the quality and yield of apple fruits. Therefore, it is urgent to cultivate new varieties of apple that are resistant to stress. *M. baccata* (L.) Borkh. has strong adaptability to the environment and good cold resistance, making it an important apple rootstock. In this study, a new WRKY TF gene, *MbWRKY46*, was isolated and cloned from *M. baccata* (L.) Borkh. and transformed into *Arabidopsis* to obtain transgenic plants. Its functions in cold and drought resistance were characterized. The research results establish a theoretical basis for cultivating new varieties with stress resistance, provide genetic improvement target genes for genetic engineering breeding technology, and further enrich the information about the WRKY TF family in stress resistance function.

## 2. Results

### 2.1. Bioinformatics Analysis of MbWRKY46

The analysis results of ProtParam (http://www.expasy.org/tools/protparam.html (accessed on 4 July 2023) can be seen from Appendix A that MbWRKY46 encoded 355 amino acids (aa) and the Open reading frame (ORF) was 1068 bp. The predicted theoretical pI of MbWRKY46 protein was 5.55, the predicted molecular weight was 39.76 kDa, and the average hydrophilicity coefficient was −0.824, so it was a hydrophilic protein. The protein was composed of 20 aa, with the highest content of 5 being Ser (13.8%), Leu (7.9%), Asp (6.8%), Glu (6.8%), and Lys (6.2%). The underlined part represents the conserved domain (WRKYGQK) unique to WRKY TF, and the highlighted part showed a C2H2 type zinc finger structure, indicating that MbWRKY46 was a member of the WRKY family in Class II (Appendix A).

### 2.2. Genetic Analysis of MbWRKY46 Gene

When comparing the 10 protein sequences with high homology to MbWRKY46 found through similarity analysis in NCBI through DNAMAN, it was found that they all contained the same conserved domain (WRKYGKK) and C2H2 zinc finger domain (Figure 1A), indicating that MbWRKY46 belonged to the WRKY TF family. After phylogenetic tree analysis, it was found that MbWRKY46 was the closest and most homologous to MdWRKY46 (*Malus domestica*, RXH78129.1). The other nine protein sequences were AtWRKY46 (*Arabidopsis thaliana*, NP_182163.1), MsWRKY53 (*Malus sylvestris*, XP_050151947.1), PmWRKY53 (*Prunus mume*, XP_008239051.1), FvWRKY27 (*Fragaria vesca*, APP13922.1). ToWRKY46 (*Trema orientale*, PON82461.1), MaWRKY46 (*Melia azedarach*, KAJ4711918.1), GmWRKY41 (*Glycine max*, XP_003525349.1), QlWRKY53 (*Quercus lobata*, XP_030951908.1), and DzWRKY42 (*Durio zibethinus*, XP_022737466.1) (Figure 1B).

### 2.3. Prediction of Protein Secondary Structure and Tertiary Structure of MbWRKY46

After Protein secondary structure prediction of the MbWRKY46 protein, it was found that the content of alpha helix, extended strand, beta turn, and random coil in the protein was 20.85%, 10.14%, 3.38%, and 65.63%, respectively (Figure 2A). After SMART Program analysis, it was found that the predicted protein had a unique domain in the WRKY family (Figure 2B). After the tertiary structure prediction of MbWRKY46 protein on the SWISS-MODEL website, it was found that this result was consistent with the prediction result of Protein secondary structure and conformed to the tertiary structure characteristics of WRKY family proteins (Figure 2C).

### 2.4. Localization of MbWRKY46 Protein in the Nucleus

The fluorescence distribution of MbWRKY46-GFP observed under the confocal microscopy is shown in Figure 3. There was 35S: GFP green fluorescence distribution in the whole cytoplasm (Figure 3B), while the green fluorescence of MbWRKY46-GFP fusion protein can only be observed in the nucleus (Figure 3E). The location of the nucleus was confirmed by 6-diamino-2-phenylindole (DAPI) staining (Figure 3F). Therefore, it can be proved that MbWRKY46 protein was a nucleo protein.

### 2.5. Analysis of the Expression Level of MbWRKY46 Gene

In order to analyze the expression of *MbWRKY46* in general, qPCR detection was performed on *MbWRKY46* in the roots, stems, new leaves, and mature leaves of *M. baccata*. The results showed that the expression level of MbWRKY 46 was higher in mature leaves and roots but lower in new leaves and stems. The expression level in the root can reach 1.22 times that in mature leaves, 1.89 times that in new leaves, and 3.95 times that in stems, indicating that the expression of *MbWRKY46* in *M. baccata* had tissue specificity. These results can be seen in Figure 4A. Different stress treatments (low temperature, high temperature, high salt, drought, ABA treatment) were applied to *M. baccata*. Due to the higher expression level of *MbWRKY46* in roots and mature leaves, RNA was extracted from these two tissues and reverse transcribed. The sampling time was 0 h, 2 h, 4 h, 6 h, 8 h, and 12 h, respectively. From Figure 4B,C, it can be seen that the expression level of *MbWRKY46* in both tissues increases first and then decreases. Under different treatments, the time at which genes reached their peak varies. In mature leaves, the expression level of *MbWRKY46* was highest after 4 h of low temperature treatment, while it reached its peak at 6 h under other stress conditions. In the root, the fastest treatment to reach the highest expression level of *MbWRKY46* was the ABA treatment, which lasted for 4 h. The time to reach the highest expression level under other stress conditions was 6 h. Since the highest expression levels of *MbWRKY46* under low temperature and drought conditions were significantly higher than those under other stress conditions in these two tissues, further exploration will be conducted on how *MbWRKY46* played a role in plant tolerance response to low temperature and drought.

### 2.6. Tolerance of Transgenic A. thaliana to Low Temperature Stress

After qPCR analysis of the expression level of *MbWRKY46* in 8 transgenic *A. thaliana*, it was found that *MbWRKY46* had a high expression level in 3 strains (S2, S5, and S7) (Figure 5A). Figure 5B showed the phenotypic changes of *A. thaliana* before and after low temperature treatment. *A. thaliana* with similar growth trends showed varying degrees of wilting after 12 h of low temperature (−4 °C) treatment. After 7 d of growth in a normal environment, the recovery was significantly different. WT and UL plants turned yellow and wilted extensively, while the S2, S5, and S7 strains had less leaf damage (Figure 5B).

After calculating the survival rates of each strain before and after being subjected to stress, it was found that the survival rates of all strains were almost the same without treatment, about 90%. After low-temperature treatment, the average survival rates of WT and UL were only 24.8% and 23.4%, respectively, while the survival rates of transgenic *A. thaliana* were still relatively high, at 83.2% (S2), 81.8% (S5), and 80.1% (S7), respectively (Figure 5C). Therefore, it can be preliminarily concluded that transforming *MbWRKY46* can improve the survival ability of *A. thaliana* under cold conditions.

In order to gain a clearer understanding of the cold tolerance of transgenic *A. thaliana*, the relevant physiological indicators of each strain before and after low-temperature treatment were measured and compared. The comparison results were shown in Figure 6, and there was no significant difference in the indicators of all *A. thaliana* under no stress. After growing for 12 h under cold conditions, there were significant differences in these indicators between WT, UL, and transgenic strains. The chlorophyll content in all plants decreased, but more so in WT and UL. Other indicators increased significantly. Except that the content of Malondialdehyde (MDA) was higher in WT and UL lines, the activity of Superoxide dismutase (SOD), peroxidase (POD), Catalase (CAT), and proline content were higher in transgenic lines. Therefore, it can be further explained that the transformation into *MbWRKY46* significantly enhanced the ability of *A. thaliana* to clear reactive oxygen species (ROS) and its resistance to low temperature.

### 2.7. Tolerance of Transgenic A. thaliana to Low Temperature Stress

Figure 7A showed the phenotypic changes of *A. thaliana* before and after drought treatment. After 10 d of drought stress treatment, the stems and leaves of *A. thaliana* with similar growth vigor drooped and wilted significantly. After 7 d of growth in a normal environment, the growth of WT and UL plants did not improve, while S2, S5, and S7 plants gradually resumed growth.

After calculating the survival rates of each strain before and after being subjected to stress, it was found that the survival rates of all strains were almost the same without treatment (about 90%). After low-temperature treatment, the average survival rates of WT and UL were only 14.4% and 13.5%, respectively, while the survival rates of transgenic *A. thaliana* were still relatively high (80.6% (S2), 81.8% (S5), and 80.2% (S7), respectively). Therefore, it can be preliminarily concluded that transforming *MbWRKY46* can improve the survival ability of *A. thaliana* under drought conditions (Figure 7B).

In order to further understand the drought resistance of transgenic *A. thaliana*, the relevant physiological indicators of each strain before and after drought treatment were measured and compared. The comparison results were shown in Figure 8, and there was no significant difference in the indicators of all *A. thaliana* plants under no stress. After 10 d of drought treatment, there were significant differences in these indicators between the WT, UL, and transgenic strains. The change results were consistent with the changes after low temperature stress. Except for the decrease of chlorophyll content, other indicators were significantly increased, and except that the content of MDA was higher in WT and UL lines, other indicators were higher in overexpression lines. Therefore, it can be further explained that transferring to *MbWRKY46* can reduce the threat of drought to *Arabidopsis*.

### 2.8. Expression Analysis of Stress-Related Downstream Genes in MbWRKY46-OE A.thaliana

WRKY TFs can regulate the expression of downstream genes by specifically binding to the W-box of downstream target gene promoters. The expression level analysis of genes related to low temperature and stress downstream of *MbWRKY46* was shown in Figure 9. Before stress treatment, the expression levels of *AtKIN1*, *AtRD29A*, *AtCOR47A*, *AtDREB2A*, *AtRD29B*, and *AtERD10* in WT, UL, and *MbWRKY46*-OE *A. thaliana* lines were basically at the same low level. After stress, the expression levels of these genes increased in all strains, but the increase was more significant in overexpressed strains. It indicated that overexpression of *MbWRKY46* can upregulate the expression levels of downstream genes related to cold stress, improving plant resistance to cold and drought.

## 3. Discussion

Plants have evolved various mechanisms to cope with stress in order to survive under constantly changing environmental conditions, including at the physiological, morphological, cellular, and molecular levels [21]. As an important part of the regulatory network, TFs can combine with cis-acting elements in the promoter region of target genes in the signal transduction pathway to regulate gene expression [22]. Many TF genes are widely used as potential candidate genes in plant breeding. There are many members of the WRKY-TFs family, and it has been confirmed that WRKY-TFs play a crucial role in regulating plant growth and development [23,24]. In addition, they can help plants resist adverse environments [25]. However, at present, the research on WRKY TFs mainly focuses on Poaceae plants, and the research on apples, especially WRKY TFs in *M. baccata*, is less.

This study used *M. baccata* as the experimental material to obtain a new *MbWRKY46* gene through homologous cloning technology and conducted bioinformatics analysis to predict the biological function of the target protein. After analysis, MbWRKY46 encoded 355 aa, and the ORF was 1068 bp. The theoretical pI of the predicted protein was 5.55, the predicted molecular weight was 39.76 kDa, and the average hydrophilicity coefficient was −0.824; as such, it was a hydrophilic protein. This protein was composed of 20 aa, with Ser, Leu, Asp, Glu, and Lys being the five proteins with the highest content. This aa sequence contained the WRKYGQK domain and a C2H2 type zinc finger structure, which conformed to the characteristics of WRKY TF class II. The subcellular localization results indicated that the MbWRKY46 protein was localized in the nucleus. Through sequence alignment and genetic analysis, MbWRKY46 had the highest homology with MdWRKY46, which further indicated that MbWRKY46 was a member of the WRKY TF family and can specifically combine with W-box action elements to play a regulatory role. It has been proved that VvWRKY28 from Class II WRKY TFs family can transgene *Arabidopsis* to resist cold and high salt [26]. Therefore, we speculated that *MbWRKY46* was involved in plant response to abiotic stress.

In order to investigate how the *MbWRKY46* gene was regulated in plant stress response, the expression level of *MbWRKY46* in the roots, stems, young leaves, and mature leaves of the *M. baccata* was first detected using qPCR. It was found that the gene was more easily expressed in mature leaves and roots, as shown in Figure 4A. Therefore, by extracting mRNA from mature leaves and roots for the next step of research, as shown in Figure 4B,C, it can be seen that, after low temperature, high temperature, high salt, drought, and ABA treatment, the expression level of *MbWRKY46* was the highest in mature leaves after 4 h of low temperature treatment, while the peak time under other stress conditions was 6 h. In the root, the fastest treatment to reach the highest expression level of *MbWRKY46* was ABA treatment, which lasted for 4 h. The time to reach the highest expression level under other stress conditions was 6 h, indicating that the expression of *MbWRKY46* had tissue and spatial specificity. The detection results also indicated that *MbWRKY46* was more sensitive to the stimulation of cold and drought in these two tissues. Therefore, we inferred that this gene was involved in regulating the response of plants to low temperature and drought stress, and this regulatory process was mediated by ABA.

To verify our hypothesis, *MbWRKY46* was transferred into *Arabidopsis* to analyze its expression under low temperature and drought conditions to explore the impact of *MbWRKY46* expression on cold and drought resistance. WT and UL strains, as well as three transgenic strains (S2, S5, and S7), were subjected to cold and drought treatment to record phenotypic changes in *Arabidopsis* and calculated survival rates before and after stress treatment. It was found that after stress treatment, the damage to WT and UL was more severe than that of the transgenic lines. After one week of recovery under normal conditions, the phenotype of the transgenic strains basically returned to normal and the survival rate was still high, while WT and UL wilted more severely and most of them died. After being threatened by adverse environmental conditions, many related physiological indicators of plants will undergo corresponding changes in order to adapt to harsh environments. When plants are subjected to environmental stress, ROS in their cells increases, and excessive ROS can have an impact on plant metabolism, seriously damaging plant health [27]. Therefore, it is necessary to use antioxidant mechanisms to regulate the balance of ROS in cells and reduce the threat of the environment to plants [28]. SOD is an important antioxidant enzyme in organisms, which is the main substance for clearing reactive oxygen species [29]. Its activity can reflect the degree of damage to plants. CAT can promote the decomposition of H_2_O_2_, accelerate the clearance of excess H_2_O_2_ within cells, and reduce the damage of H_2_O_2_ to cells [30]. POD has high activity and can catalyze H_2_O_2_, thereby reducing the content of reactive oxygen species [31]. Proline, as an important osmoregulation substance, plays an important role in maintaining the osmotic balance inside and outside cells [32]. However, excessive proline can also have a negative impact on plant growth [33]. Stress can affect photosynthesis, not only affecting the synthesis of chlorophyll, but also accelerating the decomposition of synthesized chlorophyll, causing leaves to turn yellow [34]. When the growth and development of plants are threatened, the peroxidation reaction of tissue or organ membrane lipids will produce MDA [35], so the content of MDA can also be used to determine the degree of damage to plants. After experiencing low temperature and drought stress, the chlorophyll content in all plants decreased, but more in WT and UL. Other indicators increased significantly. Except that the content of MDA was higher in WT and UL lines, the activities of SOD, POD, CAT, and proline content were higher in transgenic strains. The overexpression of *MbWRKY46* can enhance the scavenging ability of *Arabidopsis* to ROS, thereby enhancing its resistance to stress.

TFs are regulatory factors for gene expression, and WRKY TF regulates the transcription of target genes by directly binding to downstream gene promoter elements, playing an important role in the ABA signal transduction pathway. Research has found that ABA can positively induce the expression of *MdWRKY31*, making plants more sensitive to ABA stimulation [36]. After receiving the ABA signal, *WRKY46* binds to the promoter of the target gene *ABI5*, inhibiting the activities of *ABI5* [16]. CBF protein is also an important stress protein. WRKY can combine with cold binding factor/dehydration response element (*CBF*/*DREB*) and activate many other downstream genes, causing changes in the level of cryoprotectant and sugar metabolism, and affecting growth and cell expansion [37,38]. Many research results have proved that CBF genes in many plants respond to cold, drought, and other stimuli [39]. After analyzing the relative expression levels of several genes (*AtKIN1*, *AtRD29A*, *AtCOR47A*, *AtDREB2A*, *AtRD29B*, and *AtERD10*) located downstream of *MbWRKY46* and related to low temperature and drought stress (Figure 10), it was found that the relative expression levels of these genes in WT, UL, and overexpressing lines were almost at the same low level without stress treatment. After stress treatment, the expression levels of these genes increased, and the increase was more significant in transgenic strains; *MbWRKY46* was overexpressed after feeling the stimulation of cold and drought, and then activated the expression of downstream target genes through the CBF pathway or ABA dependent pathway, thereby improving the cold and drought resistance of transgenic plants.

Based on the above results and previous research findings, we have established a model to describe the possible pathways through which *MbWRKY46* regulates plant responses to low temperature and high salinity. *MbWRKY46* is overexpressed after being stimulated by cold and high salt, and then binds to CBF/DREB or participates in ABA synthesis to transmit signals downstream to target genes related to cold and drought stress, thereby activating their expression, enabling plants to have stronger survival ability under cold or drought conditions.

## 4. Materials and Methods

### 4.1. Cultivation and Treatment of M. baccata

First, seeds of *M. baccata* were treated by accelerating germination. The seeds were sown in MS medium containing sugar (30 g/L) and agar (7.5 g/L) [40]. After germination, they were transferred to a new MS medium containing indole butyric acid (IBA) and cytokinin (6-BA) for rapid propagation (0.6 mg/L IBA + 0.6 mg/L 6-BA). After two weeks, we transferred the robust tissue culture seedlings to an MS rooting medium (1.2 mg/L IBA + 0.6 mg/L6-BA) until they took root [41]. After developing developed root systems, 30 tissue culture seedlings were selected for hydroponics, and the Hoagland nutrient solution was replaced every 3–4 days. The environment in the tissue culture room was kept constant (at a temperature of around 25 °C and a relative humidity of 80–85%) [42]. After growing seven to nine fully unfolded true leaves, these well grown hydroponic seedlings were evenly divided into six groups for different stress treatments. The first group did not undergo any treatment and sampled the roots, stems, and leaves (new and mature leaves) of this group of seedlings. The remaining five groups were placed under the following five stress conditions: (1) the hydroponic seedlings were placed in 4 °C light incubator for low-temperature stress treatment; (2) replaced the hydroponic solution with 200 mM NaCl, which resulted in tissue cultured seedlings being subjected to salt stress in a high salt environment; (3) replaced the hydroponic solution with Hoagland nutrient solution with a PEG6000 concentration of 20% in order to simulate a dry environment; (4) placed the hydroponic seedlings in a light incubator at a temperature of 37 °C for high-temperature stress treatment; (5) increased the concentration of ABA in the hydroponic solution to 50 μM subjected water cultured seedlings to ABA stress treatment. At the 0 h, 2 h, 4 h, 6 h, 8 h, and 12 h of treatment, samples of new leaves and roots of hydroponic seedlings under these five treatments were taken and immediately placed in liquid nitrogen, and the samples were stored at −80 °C.

### 4.2. Obtaining the Full Length of MbWRKY46 Gene

Samples were taken from the roots, stems, new leaves, and mature leaves of the *M. baccata*. The total RNA of the samples was extracted using the OminiPlant RNA kit (Conway Collection, Beijing, China), and the first strand of cDNA was synthesized using the reverse transcription kit (TransGen, Beijing, China, Article number: AE311-02). Using the CDS region of *MdWRKY46* as a reference sequence, this sequence was derived from NCBI (https://www.ncbi.nlm.nih.gov/ (accessed on 25 June 2023)). Two pairs of specific primers *MdWRKY*46-F/R (Appendix A) were designed using Primer 5.0 software. Using synthesized cDNA as a template, PCR amplification was performed on the target fragment to obtain the full length of the target gene. The PCR reaction system and reaction conditions were shown in Appendix A [43,44].

After the PCR product was purified by gel, using *pEASY*^®^-T5 cloning vector (TransGen, Beijing, China, Article number: CT501) and *Trans*1-T1 competent cells (Article number: CD501) to connect the target gene with the expression vector, and then carried out sequencing after verification (Huada Gene, Beijing, China).

### 4.3. Bioinformatics Analysis of MbWRKY46 Gene

The sequencing results of the *MbWRKY46* gene were determined by EMBOSS Needles (http://www.ebi.ac.uk/Tools/psa/emboss_needle/ (accessed on 4 July 2023)) and we translated the nucleic acid sequence into amino acid sequence using DNAMAN5.2. BLAST (https://blast.ncbi.nlm.nih.gov/Blast.cgi (accessed on 4 July 2023)) was performed on the amino acid sequence of *MbWRKY46* in NCBI (https://www.ncbi.nlm.nih.gov/ (accessed on 4 July 2023)), and 10 amino acid sequences of other species with higher homology to *MbWRKY46* were selected for similarity comparison with *MbWRKY46*. These 10 amino acid sequences were AtWRKY46 (*Arabidopsis thaliana*, NP_182163.1), MdWRKY46 (*Malus domestica*, RXH78129.1), MsWRKY53 (*Malus sylvestris*, XP_050151947.1), PmWRKY53 (*Prunus mume*, XP_008239051.1), FvWRKY27 (*Fragaria vesca*, APP13922.1), ToWRKY46 (*Trema orientale*, PON82461.1), MaWRKY46 (*Melia azedarach*, KAJ4711918.1), GmWRKY41 (*Glycine max*, XP_003525349.1), QlWRKY53 (*Quercus lobata*, XP_030951908.1), and DzWRKY42 (Durio zibethinus, XP_022737466.1). MEGA7 was used to construct an evolutionary tree. In addition, the primary, secondary, tertiary structures, and domains of MbWRKY46 protein were predicted using ExPASy (https://web.expasy.org/protparam/ (accessed on 7 July 2023)), SOPMA (https://npsa-prabi.ibcp.fr/ (accessed on 7 July 2023)), SWISS-MODEL (https://swissmodel.expasy.org/ (accessed on 7 July 2023)), and SMART (http://smart.embl-heidelberg.de/ (accessed on 7 July 2023)), respectively.

### 4.4. Subcellular Localization of MbWRKY46 Protein

The subcellular localization expression vector was selected as 35S-sGFP-pCAMBIA1300. Two restriction endonuclease sites *BamH*Ⅰ and *Sal*Ⅰ were selected on the vector, and upstream and downstream primers containing these two restriction endonuclease sites were designed (*MdWRKY46*-2F/R) (Appendix A). The target protein MbWRKY46 and plasmid vector were digested by double enzymes through *Xba*I and *kpn*I restricted Endonuclease, and then the target fragment of MbWRKY46 was inserted into the plasmid vector to obtain the transient expression vector of MbWRKY46. The gold powder containing the plasmid MbWRKY46 and the control plasmid was injected into the epidermal cells of onion using the particle bombardment method, and then the location of the target protein was observed using the confocal microscopy (LSM 510 Meta, Zeiss, Wetzlar, Germany).

### 4.5. Real Time Fluorescence Quantitative PCR Analysis of MbWRKY46 Gene

Designed a specific primer *MbWRKY46*-qF/qR (Appendix A) based on the conserved domain of *MbWRKY46*, using the cDNA obtained in 2.2 as the template and *Actin* as the internal reference gene. Used qPCR to detect the expression level of *MbWRKY46* in different tissues (roots, stems, young leaves, and mature leaves) under stress and normal conditions, and performed one-way ANOVA. The obtained data were used 2^−ΔΔCT^ method for analysis [45]. The reaction system and conditions were the same as those in 2.2 (Appendix A).

### 4.6. Obtaining Transgenic Plants

The pCAMBIA2300 plasmid vector was digested with *Xba*I and *kpn*I restricted endonuclease to obtain a linearized vector, and the target fragment of *MbWRKY46* was inserted between the *Xba*I and *kpn*I restriction sites of the vector to obtain the overexpression vector pCAMBIA2300-*MbWRKY46*. The homologous primer used was *MbWRKY46*-hF/hR, as shown in Appendix A.The vector was transferred into the GV3101 competent state for transformation of Agrobacterium tumefaciens, and then Agrobacterium tumefaciens was transferred to the Colombian Ecotype *Arabidopsis* by inflorescence mediated method [46]. After seed germination, transfered T_1_ generation seeds to MS screening medium (containing 50 mg/L kanamycin) for preliminary identification of transgenic *A. thaliana*. Performed qPCR analysis on T_2_ generation plants for final determination, and seted WT and UL as control groups. Used the T_3_ generation transgenic plants as experimental materials for further processing and analysis [47].

### 4.7. Determination of Relevant Physiological Indicators

Transfer the control group and T_3_ generation transgenic plants to a nutrient bowl (nutrient soil:vermiculite = 2:1) [48]. After the plants grow vigorously (about 30 days), they are divided into two groups, each with 20 plants, and subjected to different stress treatments. The first group was placed in a −4 °C incubator for low temperature stress treatment (12 h) [49], and the second group stopped watering for drought treatment (10 d). After treatment, they were placed back under normal conditions for a week to recover. Record the phenotypic changes of each plant before and after treatment, measure the changes in relevant physiological indicators before and after treatment, and calculate the survival rate of each plant line. The measurement methods for relevant physiological indicators are as follows: Chlorophyll was extracted by extraction method [50]. SOD activity was measured by nitrogen blue tetrazole photo reduction method [51], POD activity was measured by Guaiacol method [52], MDA content was measured by spectrophotometer colorimetry [53], CAT activity was measured by ultraviolet absorption method [54], and proline content was extracted and measured by sulfosalicylic acid method [55].

### 4.8. Expression Analysis of Downstream Genes of MbWRKY46

The RNA of WT, UL, and transgenic *A. thaliana* before and after stress treatment was extracted according to the method in 2.2, and it was reverse transcribed into the first strand cDNA as an amplification template. The stress-related target genes located downstream of *MbWRKY46* were qPCR detected (refered to 2.2) to analyze the relative expression levels of these genes. The specific primers used were shown in Appendix A.

### 4.9. Statistical Analysis

Single factor analysis of variance is a statistical method used to compare whether there is a significant difference in the mean values between three or more groups. In this study, the average values of all indicators were obtained through Threefold repetition tests, and then the SPSS 21.0 software (IBM, Chicago, IL, USA) was used to perform a one-way ANOVA on the average values obtained. Then, the standard deviation measurement (SD) was performed on these data. The calculated significant difference (*p*-value) can be used to represent the difference between experimental results. When *p* ≤ 0.05, the *p*-value is represented by *, indicating a significant difference between the two groups of data (* *p* ≤ 0.05). When *p* ≤ 0.01, the *p*-value is represented by **, indicating a very significant difference between the two groups of data.

## 5. Conclusions

In this study, a new *WRKY* gene was isolated from *M. baccata* and was given the name *MbWRKY46*. *MbWRKY46* protein is a nucleo protein and had the highest homology and closest genetic relationship with *MdWRKY46*. Cold, heat, high salinity, drought, and ABA can all stimulate overexpression of *MdWRKY46*, especially cold and drought. Cold and drought stress on *MdWRKY46*-OE *A. thaliana* can cause changes in the physiological indicators of the plant to adapt to stress. Overexpression of *MdWRKY46* activated the expression of downstream target genes, thereby improving plant cold and drought tolerance. This study demonstrated that *MdWRKY46* can improve plant tolerance to cold and high salt.

## Figures and Tables

**Figure 1 ijms-24-12468-f001:**
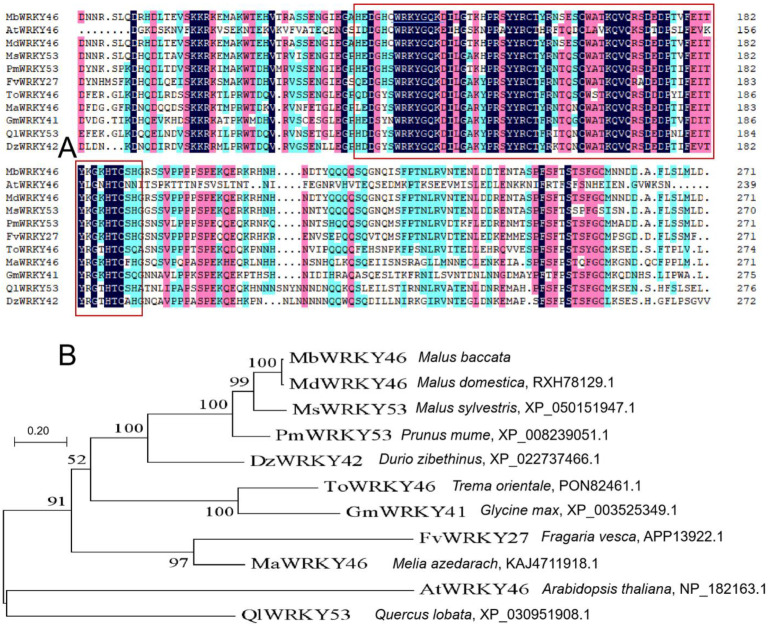
Comparison (**A**) and evolutionary tree analysis (**B**) of the amino acid sequences of MbWRKY46 and WRKY-TF proteins of Other Species. Note: The one in the red box was the conserved sequence, and the one marked with blue underline was the target protein.

**Figure 2 ijms-24-12468-f002:**
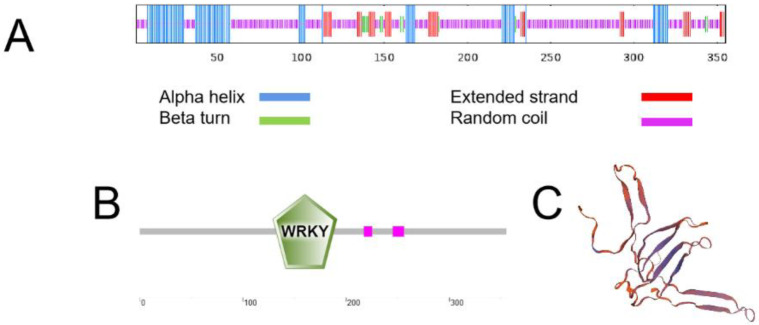
Prediction of Secondary and Tertiary Structure of MbWRKY46 Protein. (**A**) Predicted protein secondary structure; (**B**) predicted protein domains; (**C**) predicted tertiary structure.

**Figure 3 ijms-24-12468-f003:**
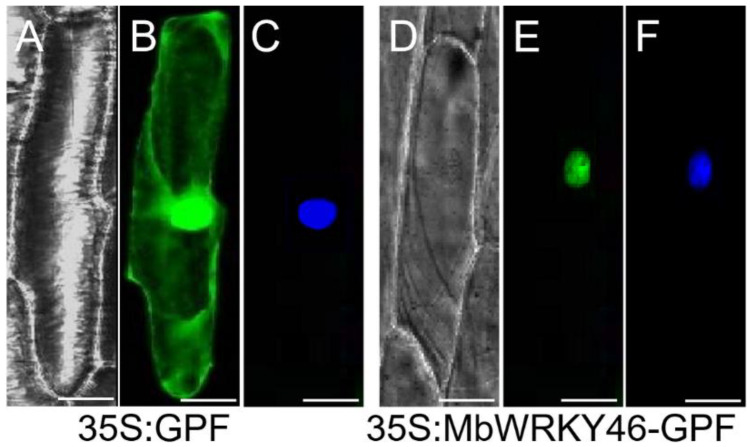
Subcellular localization of MbWRKY46 protein. They were observed under bright light (**A**,**D**) using a fluorescence microscope, a GFP signal image (**B**,**E**) and a DAPI staining image (**C**,**F**) under dark conditions. Bar = 50 μm.

**Figure 4 ijms-24-12468-f004:**
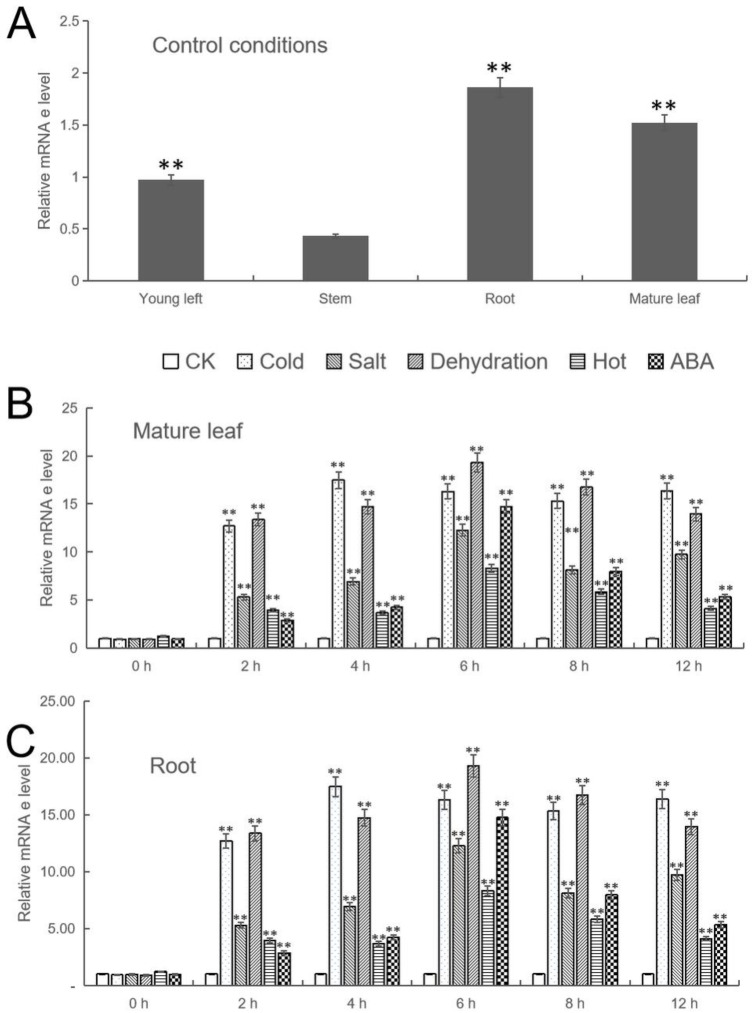
Results of qPCR analysis of *MbWRKY46*. (**A**) The expression of *MbWRKY46* in different organs of *M. baccata*; The expression changes of *MbWRKY46* in mature leaves (**B**) and roots (**C**) over time under different stress treatments. Compared with the Control (CK), the asterisks above the column indicate significant difference and extremely significant difference (**, *p* ≤ 0.01).

**Figure 5 ijms-24-12468-f005:**
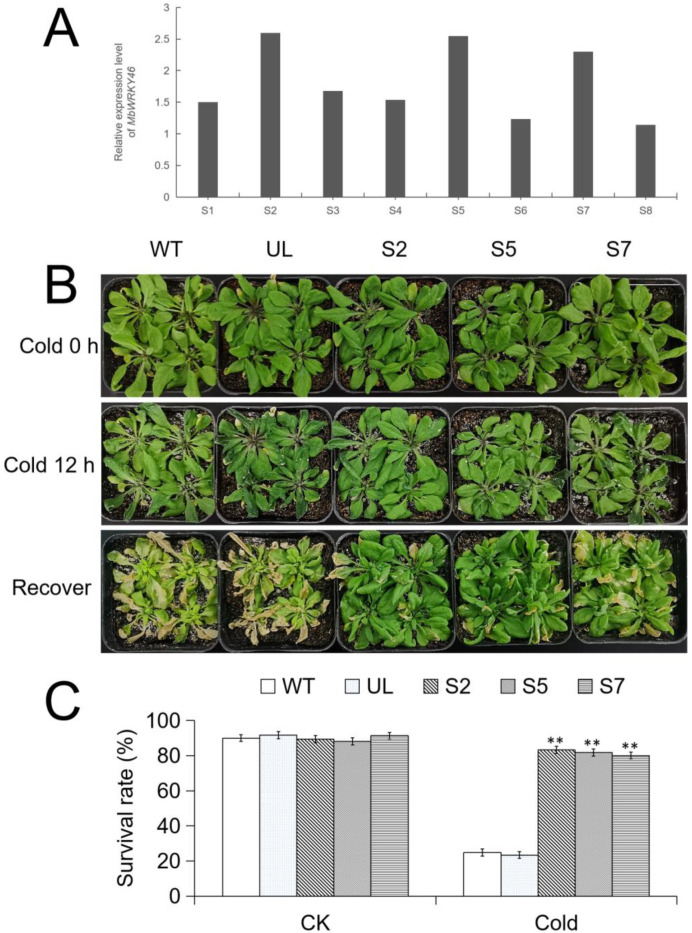
Phenotype and survival rate of *A. thaliana* under cold stress. (**A**) The relative expression of *MbWRKY46* in transgenic *A. thaliana*. Phenotypic changes (**B**) and survival rate changes (**C**) of WT, UL, S2/5/7 strains before and after treatment at −4 °C and after recovery of growth. Bar = 5 cm. (**, *p* ≤ 0.01).

**Figure 6 ijms-24-12468-f006:**
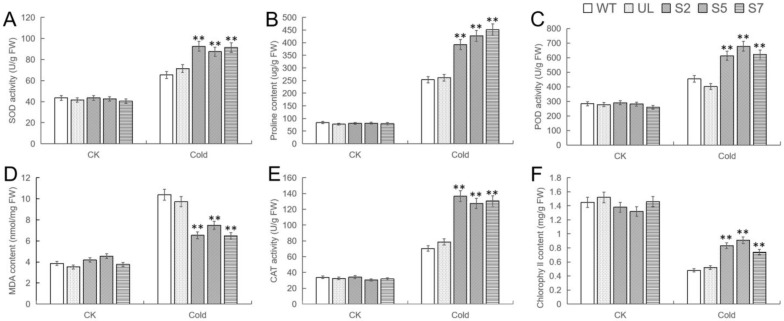
Changes in physiological and biochemical indicatorsin *MbWRKY46*-OE *A. thaliana* under cold conditions. (**A**) SOD activities, (**B**) proline content, (**C**) POD activities, (**D**) MDA content, (**E**) CAT activities, and (**F**) Chlorophyll content. (**, *p* ≤ 0.01).The control was the index in the WT. All data were the average of 3 measurements.

**Figure 7 ijms-24-12468-f007:**
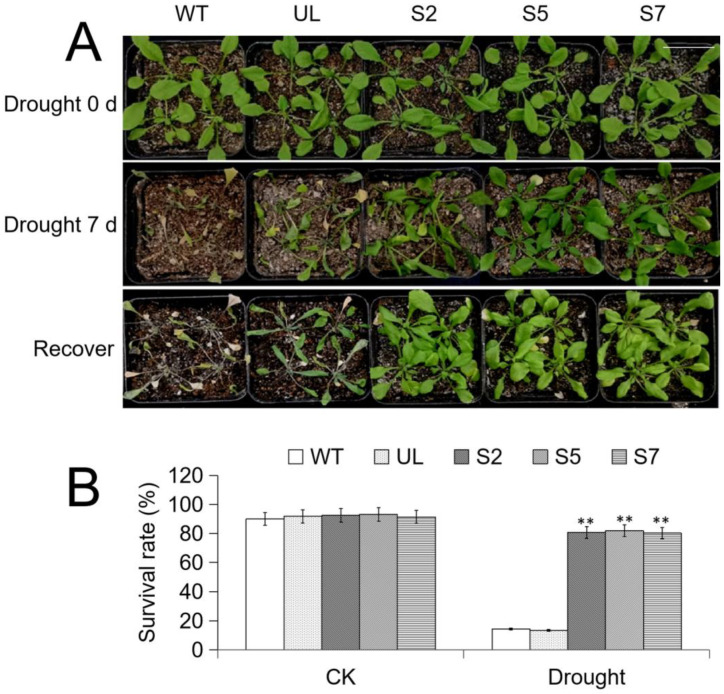
Phenotype and survival rate of *A. thaliana* under drought stress. (**A**) The relative expression of *MbWRKY46* in WT, UL and transgenic *A. thaliana*. (**B**) Phenotypic changes and survival rate changes of WT, UL, S2/5/7 strains before and after water deficiency treatment and after recovery of growth. Bar = 5 cm. (**, *p* ≤ 0.01).

**Figure 8 ijms-24-12468-f008:**
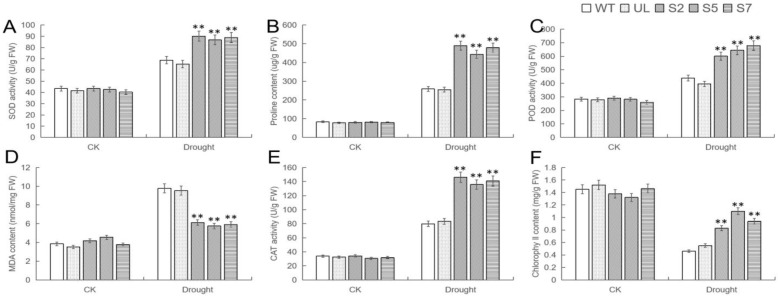
Changes in physiological and biochemical indicators in *MbWRKY46*-OE *A. thaliana* under drought conditions. (**A**) SOD activities, (**B**) proline content, (**C**) POD activities, (**D**) MDA content, (**E**) CAT activities, and (**F**) Chlorophyll content. (**, *p* ≤ 0.01). The control was the index in the WT. All data were the average of 3 measurements.

**Figure 9 ijms-24-12468-f009:**
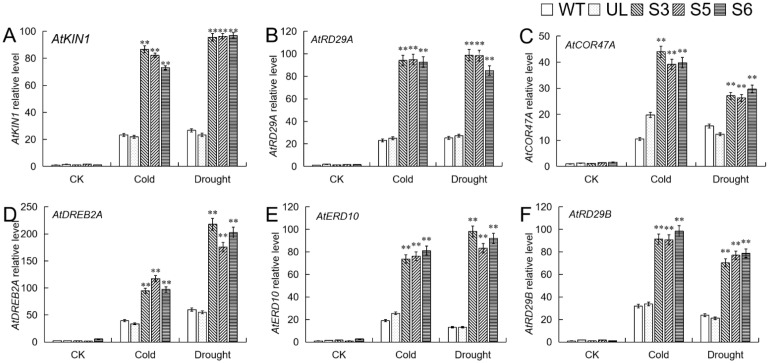
Expression of cold and drought stress related genes in *A. thaliana*. The expression levels of (**A**) *AtK1N1*, (**B**) *AtRD29A*, (**C**) *AtCOR47A*, (**D**) *AtDREB2A*, (**E**) *AtERD10* and (**F**) *AtRD29B*. (**, *p* ≤ 0.01). The control was the index in the WT. All data were the average of 3 measurements.

**Figure 10 ijms-24-12468-f010:**
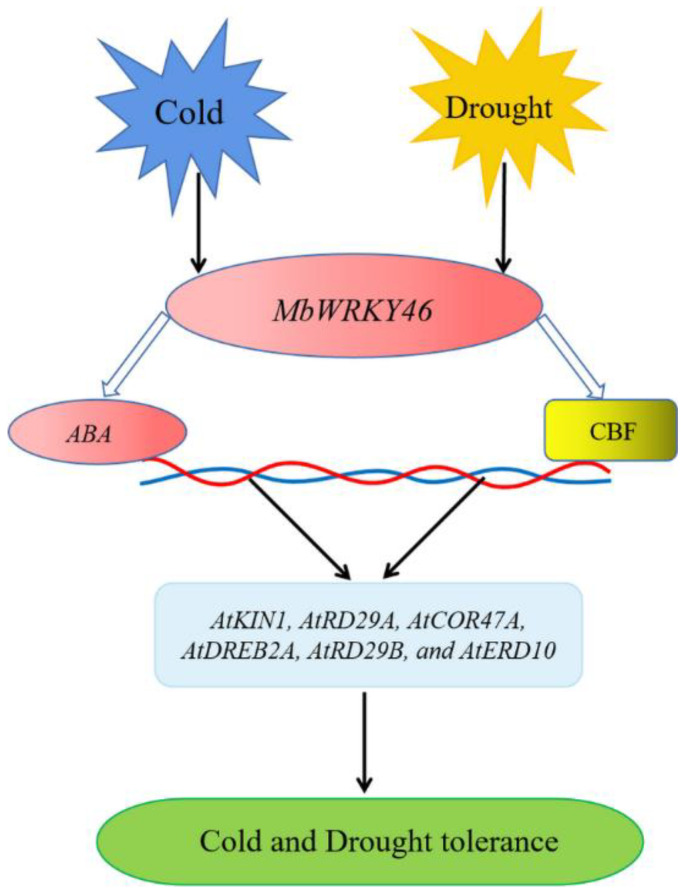
A potential model of MbMRKY46 regulating plant responses to low temperature and drought stress.

## Data Availability

The original data for this present study are available from the corresponding authors.

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
