# Peer review of "The Transcription Factor MbWRKY46 in Malus baccata (L.) Borkh Mediate Cold and Drought Stress Responses"

_ijms, 2023, doi:10.3390/ijms241512468_

Round 1

Reviewer 1 Report

Isolation and Functional Analysis of Malus baccata (L.) Borkh WRKY transcription factors MbWRKY46 is a study that meets the needs of plants to adapt to extreme climatic variations. The results of this study have certain reference value for the application of M. baccata MbWRKY46 in low-temperature and drought response, and also provide a theoretical basis for further research on its function in the future. In this study, a new WRKY gene was isolated from M. baccata, and was given the name MbWRKY46. MbWRKY46 protein was a nucleo protein, and had the highest homology and closest genetic relationship with MdWRKY46. In order to gain a clearer understanding of the cold tolerance of transgenic A. thaliana, the relevant physiological indicators of each strain before and after low-temperature treatment were measured and compared. These results were obtained by combining complex analytical methods and techniques, and encourage further research.

Author Response

Isolation and Functional Analysis of Malus baccata (L.) Borkh WRKY transcription factors MbWRKY46 is a study that meets the needs of plants to adapt to extreme climatic variations. The results of this study have certain reference value for the application of M. baccata MbWRKY46 in low-temperature and drought response, and also provide a theoretical basis for further research on its function in the future. In this study, a new WRKY gene was isolated from M. baccata, and was given the name MbWRKY46. MbWRKY46 protein was a nucleo protein, and had the highest homology and closest genetic relationship with MdWRKY46. In order to gain a clearer understanding of the cold tolerance of transgenic A. thaliana, the relevant physiological indicators of each strain before and after low-temperature treatment were measured and compared. These results were obtained by combining complex analytical methods and techniques, and encourage further research.

Response: We appreciate your feedback on our manuscript. I think they are very important, and in future research and writing, we will work harder to achieve better results.

Reviewer 2 Report

In general I thought that the writing was careguully done 

this sentence should be changed. 

It was found that this gene had higher expression was more easily expressed in mature leaves and roots, followed by new leaves, with the lowest expression level in stems.

the title is boring. state what the paper is about

The transcription factor MbWRKY46  in Malus baccata (L.) Borkh  mediate cold and drought stress responses. 

I think an interesting next step is to look for upregulation of specific enzymes and transporters that might transmit long distance signaling.  Based on the phenotypes I would look for altered expression of enzymes related to putrescine synthesis  ADC and ARGAH possibly ODC.and transport.

no comments

Author Response

In general I thought that the writing was careguully done 

We appreciate your valuable suggestions and advices on our manuscript. I think they are very helpful and important, and revisions had been made in the revised manuscript accordingly.

Here I would like to response the comments and add some explanations as follows.

this sentence should be changed. 

It was found that this gene had higher expression was more easily expressed in mature leaves and roots, followed by new leaves, with the lowest expression level in stems.

Response: Yes, we have accepted your suggestion and have modified this part.

the title is boring. state what the paper is about

The transcription factor MbWRKY46  in Malus baccata (L.) Borkh  mediate cold and drought stress responses. 

Response: Yes, we have accepted your suggestion and have modified the title.

I think an interesting next step is to look for upregulation of specific enzymes and transporters that might transmit long distance signaling.  Based on the phenotypes I would look for altered expression of enzymes related to putrescine synthesis  ADC and ARGAH possibly ODC.and transport.

Response: We believe this is a very good and meaningful suggestion. In the following research, we will identify the upregulation of specific enzymes and transporters that may transmit long-distance signals based on your suggestion, and study how the expression of enzymes related to putrescine synthesis ADC and ARGAH changes based on phenotype.

Reviewer 3 Report

The climatic changes that affect human societies primarily affect agricultural crops. Today, there is a strong need to understand how to improve the adaptation of plants to abiotic stresses related to climate change, such as excessive drought or cold. In this regard, the authors propose a work focused on the isolation and characterization of a gene (MbWRKY46) of resistance to abiotic stresses in Malus baccata cultures. The WRKY transcription factors are involved in resistance to abiotic stresses and are studied in many plants. Specifically, MbWRKY46 has been little studied in M. baccata, so this study is very innovative on this topic.

First, the authors perform a bioinformatics study to find the sequence of the gene with the relative amino acids it expresses. They then simulate the structure that the protein should have and look for the site of maximum expression (root) of M. baccata. The authors then study the tolerance of seedlings to abiotic stress in Arabidopsis models.

The manuscript is very clear and the results are well discussed, but some critical issues remain:

1) The statistical analysis is unclear. Just stating the "p" is not enough. The authors should define what statistical test they used and then revise the comments on the data.

2) Why is the standard error of the mean or the standard deviation of the values missing from Figure 5A?

3) Why is there no control group?

4) See this paper for references: Effect of Light, Temperature, Salinity, and Halopriming on Seed Germination and Seedling Growth of Hibiscus sabdariffa under Salinity Stress, DOI: 10.3390/agronomy12102491

Author Response

The climatic changes that affect human societies primarily affect agricultural crops. Today, there is a strong need to understand how to improve the adaptation of plants to abiotic stresses related to climate change, such as excessive drought or cold. In this regard, the authors propose a work focused on the isolation and characterization of a gene (MbWRKY46) of resistance to abiotic stresses in Malus baccata cultures. The WRKY transcription factors are involved in resistance to abiotic stresses and are studied in many plants. Specifically, MbWRKY46 has been little studied in M. baccata, so this study is very innovative on this topic.

First, the authors perform a bioinformatics study to find the sequence of the gene with the relative amino acids it expresses. They then simulate the structure that the protein should have and look for the site of maximum expression (root) of M. baccata. The authors then study the tolerance of seedlings to abiotic stress in Arabidopsis models.

The manuscript is very clear and the results are well discussed, but some critical issues remain:

We appreciate your valuable suggestions and advices on our manuscript. I think they are very helpful and important, and revisions had been made in the revised manuscript accordingly.

Here I would like to response the comments and add some explanations as follows.

1) The statistical analysis is unclear. Just stating the "p" is not enough. The authors should define what statistical test they used and then revise the comments on the data.

Response: Yes, we have accepted your suggestion and have supplemented this section。

2) Why is the standard error of the mean or the standard deviation of the values missing from Figure 5A?

Response: The data shown in Figure 5A showed the expression levels of MbWRKY46 in 8 transgenic Arabidopsis, these data were obtained through qPCR analysis of MbWRKY46 in these strains, with the aim of selecting the 3 lines with the highest expression levels for subsequent research. And we have made modifications to this section.

3) Why is there no control group?

Response: Figure 5A showed the expression level of MbWRKY46 in transgenic lines, with the aim of selecting the three lines with the highest expression level for subsequent research. Therefore, no control group was set up, and in other studies, the control groups were all WT and UL lines.

4) See this paper for references: Effect of Light, Temperature, Salinity, and Halopriming on Seed Germination and Seedling Growth of Hibiscus sabdariffa under Salinity Stress, DOI: 10.3390/agronomy12102491

Response: Yes, we have accepted your suggestion and have added this reference.

Round 2

Reviewer 3 Report

The authors corrected the manuscript following the suggestions of the Reviewers. In my opinion the manuscript can be accepted

Author Response

The authors corrected the manuscript following the suggestions of the Reviewers. In my opinion the manuscript can be accepted

Response:Thank you for your valuable suggestions. We believed they were very useful and would work even harder in future research to obtain better research results.